

# First report of bats (Mammalia: Chiroptera) from the Gray Fossil Site (late Miocene or early Pliocene), Tennessee, USA

Nicholas J. Czaplewski

Section of Vertebrate Paleontology, Oklahoma Museum of Natural History, Norman, OK,
United States of America

## ABSTRACT

Thousands of vertebrate fossils have been recovered from the Gray Fossil Site, Tennessee, dating to the Miocene-Pliocene boundary. Among these are but eight specimens of bats representing two different taxa referable to the family Vespertilionidae. Comparison of the fossils with Neogene and Quaternary bats reveals that seven of the eight specimens pertain to a species of *Eptesicus* that cannot be distinguished from recent North American *Eptesicus fuscus*. The remaining specimen, a horizontal ramus with m3, is from a smaller vespertilionid bat that cannot confidently be assigned to a genus. Although many vespertilionid genera can be excluded through comparisons, and many extinct named taxa cannot be compared due to nonequivalence of preserved skeletal elements, the second taxon shows morphological similarities to small-bodied taxa with three lower premolar alveoli, three distinct m3 talonid cusps, and m3 postcristid showing the myotodont condition. It resembles especially *Nycticeius humeralis* and small species of *Eptesicus*. *Eptesicus* cf. *E. fuscus* potentially inhabited eastern North America continuously since the late Hemphillian land mammal age, when other evidence from the Gray Fossil Site indicates the presence in the southern Appalachian Mountains of a warm, subtropical, oak-hickory-conifer forest having autochthonous North American as well as allochthonous biogeographical ties to eastern Asia and tropical-subtropical Middle America.

## INTRODUCTION

The Gray Fossil Site (GFS) is a small but significant late Miocene-early Pliocene vertebrate fossil locality occurring within the Valley and Ridge Zone of the Appalachian Mountains physiographic province. In this region the Valley and Ridge Zone exhibits Paleozoic rocks distorted by numerous broad folds and synclinoria that have also been cut by thrust faulting. The Paleozoic rocks include limestone, dolomite, and marble, providing soluble rocks of the Valley and Ridge karst region, one of the most extensive and important karstic regions in the United States (*Middleton & Waltham, 1992*). Prior to the discovery of GFS in 2000, karstic deposits with Neogene vertebrate fossils including bats were unknown in Tennessee (*Corgan & Breitburg, 1996*). GFS is located about 15 km NW of Johnson

Corresponding author
Nicholas J. Czaplewski,
nczaplewski@ou.edu

City, Tennessee, near the small town of Gray in northeasternmost Tennessee. GFS is situated in a small valley and reflects a depositional situation that has been interpreted as multiple sediment-filled sinkholes representing differing ages (at least Paleocene-Eocene and Miocene-Pliocene), and associated with ponds and small streams (*Shunk, 2011*; *Whitelaw, Shunk & Liutkus, 2011*; *Zobaa et al., 2011*). One of the sub-basins, GFS-2, has yielded mammalian fossils suggestive of a late Miocene or early Pliocene age, reflecting the late Hemphillian (Hh4) North American Land Mammal Age (*Wallace & Wang, 2004*; *Schubert & Mead, 2011*; *Mead et al., 2012*; *Wallace e al., 2014*). The diverse paleobiota of GFS-2 includes plant fossils, charcoalified wood, spores, and pollen that indicate a surrounding open-to-dense deciduous forest dominated by oak, hickory, conifer, and vines (including one, *Sinomenium*, having subtropical-tropical Asian affinities), and subject to occasional drought and fires and browsing by large mammalian herbivores (*Liu & Jacques, 2010*; *Zavada, 2011*; *Ochoa-Lozano & Liu, 2011*; *Ochoa et al., 2012*). Non-mammalian vertebrates recovered include bony fishes, salamanders, aquatic turtles, *Alligator*, snakes, and a beaded lizard, *Heloderma* (*Mead et al., 2012*). Preservation of biological remains is excellent and even includes abdominal contents and eggs/oocysts of internal parasites of some large mammals (*McConnell & Zavada, 2013*). Remains of *Alligator* suggest a warmer climate than today in this region, with annual low temperatures probably above 5.5 °C (*Shunk, 2011*). GFS is one of extremely few Neogene vertebrate faunas in interior eastern North America (*Janis, Gunnell & Uhen, 2008*). Certain members of the GFS fauna help to provide evidence for the relationships between paleontological events and intercontinental connections between eastern North America and Eurasia in the late Neogene (*Wallace & Wang, 2004*; *Mead et al., 2012*; *Doby & Wallace, 2014*). Although these members of the GFS fauna have been studied previously, specimens of bats have been slow to accumulate. This paper describes the first few specimens of bats yet uncovered at the GFS.

## METHODS

The first eight fossils of bats recovered from sedimentary deposits at the Gray Fossil Site by the staff and affiliates of the Gray Fossil Site and East Tennessee State University Museum of Natural History as of 2016 were graciously loaned to the author for study. The fossils were identified by comparison with casts of Neogene North American fossil bats and with skeletal material of recent bats in the Oklahoma Museum of Natural History. Occlusal terminology of the teeth and humeral terminology follow *Czaplewski, Morgan & McLeod (2008)*; terminology for anatomical features of a petrosal bone follows *Staněk (1933)*, *Henson (1970)* and *Giannini, Wible & Simmons (2006)*. Capital letters indicate upper teeth, and lower case letters indicate lower teeth. Measurements of specimens were made at 10X or 20X using an ocular micrometer on an Olympus SZX9 stereomicroscope. Abbreviations: apl, anteroposterior length; ETMNH, East Tennessee Museum of Natural History; GFS, Gray Fossil Site.

# SYSTEMATIC PALEONTOLOGY

Order Chiroptera *Blumenbach, 1779–1780*
Family Vespertilionidae *Gray, 1821*
Subfamily Vespertilioninae *Gray, 1821*
Tribe Eptesicini *Volleth & Heller, 1994*
Genus *Eptesicus Rafinesque, 1820*
*Eptesicus* cf. *E. fuscus Palisot de Beauvois, 1796*
(Figs. 1 and 2)

 Material:  East Tennessee State University Museum of Natural History (ETMNH) specimen number 9714, right M1; ETMNH 19286, left edentulous dentary fragment with partial alveoli for p4-m3; ETMNH 19287, left partial dentary with m2 and condyloid and angular processes; ETMNH 19288, right m2 possibly associated with 19287; ETMNH 9654, partial right petrosal; ETMNH 14022, left distal humerus; ETMNH 9755, left proximal radius.

 Measurements (in mm): ETMNH 9714, M1 anteroposterior length (apl), 1.8; transverse width, 2.05. ETMNH 19286 depth of dentary beneath posterior alveolus of m1, 1.7. ETMNH 19288, right m2 apl, 1.8; trigonid width, 1.1; talonid width, 1.2. ETMNH 19287 left m2 apl, 1.9; trigonid width, 1.1; talonid width, 1.2. ETMNH 14022 distal humerus midshaft diameter, 1.5; greatest transverse distal width, 3.5; extension of trochlea beyond spinous process, 0.3; proximo-distal diameter of trochlea (in anterior view), 1.8; anteroposterior diameter of trochlea (in distal view), 2.2; anteroposterior diameter of lateral ridge of capitulum (capitular tail, in lateral view), 1.9; transverse width of medial epicondyle from medial edge (lip) of trochlea, 0.6; width of distal articular surface (trochlea to capitular tail), 2.9. ETMNH 9755 proximal radius greatest width, 2.5 mm.

## Description

The right upper molar ETMNH 9714 appears to be M1, because its length and width are more nearly equal to one another than in M2s, which are anteroposteriorly shorter and transversely wider (Figs. 1A–1C). In occlusal view, these proportions give M1 a rather squarish appearance while M2s are more rectangular. The tooth shows light wear especially on the ectoloph crests. The protocone is tall and about as high as the metacone; paracone is slightly smaller. The preprotocrista extends labially almost to the parastyle. The lingual cingulum is interrupted at the base of the protocone; it extends anteriorly to the level of the base of the paracone, and extends posteriorly around the talonlike swelling and continuing as a metacingulum nearly to the metastyle. As in other vespertilionids there is no real hypocone shelf or hypocone, although the posterolingual corner of the tooth is slightly expanded as a small talon. This talon supports a small crest ("postprotocingulum" of *Gunnell, Eiting & Geraads (2011)*) that is a steeper continuation of the less-inclined postprotocrista. The protofossa is deep. Paraloph and metaloph are absent. The protofossa/trigon basin is open posteriorly. The parastylar fovea is about half the volume of the metastylar fovea, and it has a weaker labial cingulum compared to the better developed labial cingulum of the metastylar fovea. Labially the flexus of the parastylar fovea is deeper than that of the metastylar fovea.

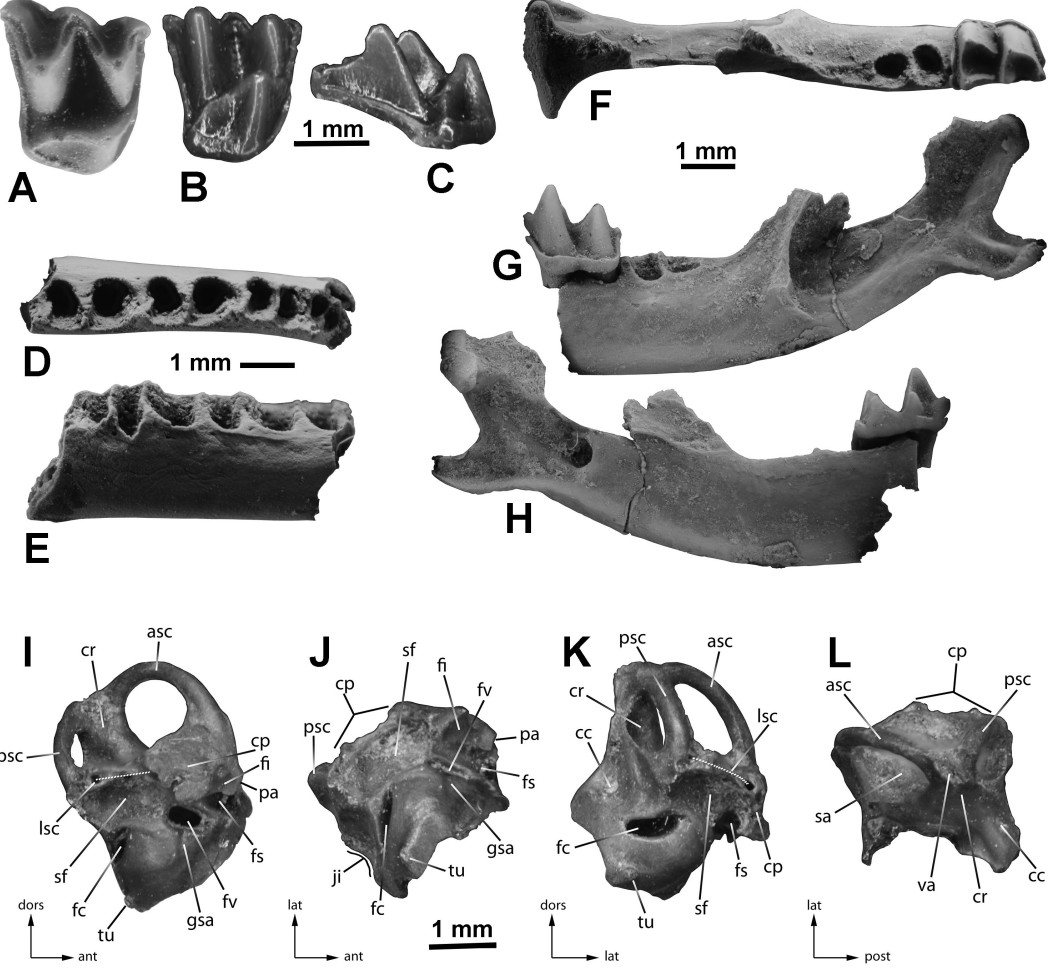

**Figure 1** **Cranial elements of *Eptesicus* cf. *E. fuscus* from the Gray Fossil Site, Tennessee.** (A–C), right M1 (ETMNH 9714) in occlusal (A, coated with ammonium chloride to reduce glare), lingual (B), and posterior (C) views. D–E, left edentulous dentary fragment with alveoli for p4-m3 (ETMNH 19286, coated) in occlusal (D), and labial (E) views. F–H, left partial dentary with m2 (ETMNH 19287, coated) in occlusal (F), labial (G), and lingual (H) views. I–L, right petrosal (ETMNH 9654) in lateral (I), ventral (J), posterior (K), and dorsal (endocranial, L) views. Abbreviations for petrosal: ant, anterior; asc, anterior semicircular canal; cc, cochlear canaliculus (plugged with sediment); cp, crista parotica (largely broken away); cr, common crus; dors, dorsal; fc, fenestra cochleae; fi, medial wall of fossa incudis; fs, facial sulcus (and semicanal for facial nerve plugged with sediment); fv, fenestra vestibuli; gsa, groove for stapedial artery; ji, jugular incisure; lat, lateral; lsc, lateral semicircular canal (dotted line indicates path of chamber exposed through broken crista parotica); pa, base of broken anterior petrosal process; post, posterior; psc, posterior semicircular canal; sa, subarcuate fossa; sf, stapedial fossa; tu, tubercle ventral to fenestra cochleae; va, opening of vestibular aqueduct (plugged with sediment).

The m2s in ETMNH 19287 (Figs. 1F–1H) and 19288 are the same size as one another, show myotodonty, have a tiny lingual cingulum restricted to the base of the trigonid valley, and a thick and prominent labial cingulum. The metaconid and entoconid are about the same height, and the entoconid bears a straight entocristid. Both m2s from GFS are the size of the m2 of *E. fuscus*. The condyloid and angular processes of the dentary are similar

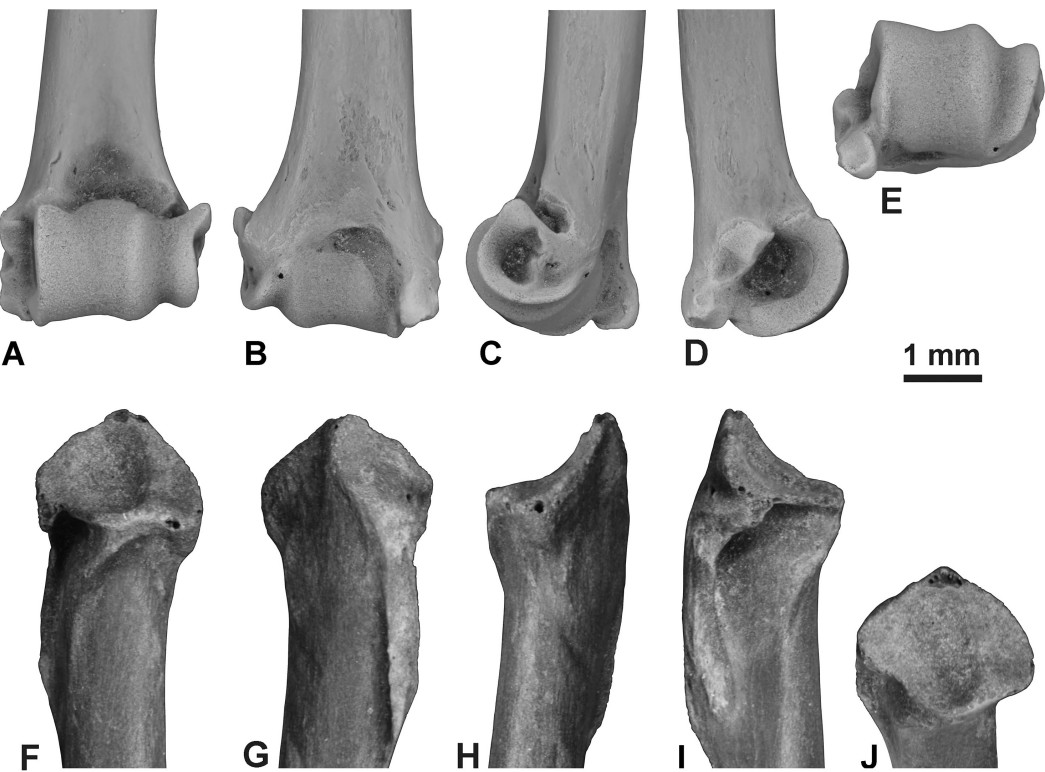

**Figure 2 Postcranial elements of *Eptesicus* cf. *E. fuscus* from the Gray Fossil Site, Tennessee.** Left distal humerus (ETMNH 14022; coated with ammonium chloride) in anterior (A), posterior (B), lateral (C), medial (D), and distal (E) views. Left proximal radius (ETMNH 9755) in anterior (F), posterior (G), lateral (H), medial (I), and proximal (J) views.

in size and shape to those in *E. fuscus*, in which some intraspecific variation occurs among the comparison specimens.

The jaw fragment ETMNH 19286 is broken through the anterior alveolus of the p4, and posteriorly it is broken through the posterior alveolus of the m3 (Figs. 1D–1E); it is from a large bat, about the same size as *E. fuscus*. The edentulous fragment includes the alveoli for a two-rooted p4. Mesial and slightly lingual to the p4 anterior alveolus is the bottom of another premolar alveolus (Fig. 1D); this probably represents the socket for the single root of a p3.

ETMNH 9654 is a damaged right petrosal missing most of the pars cochlearis and parts of the crista parotica and lateral semicircular canal (Figs. 1I–1L). The posterior semicircular canal bears a thin, small partial lamina occurring as a flange along its ventral edge, and the anterior semicircular canal bears a short flangelike lamina near either end. There is no complete lamina capping the area between the three semicircular canals as seen in petrosals of members of the Emballonuridae, Rhinolophidae, and some Molossidae (GS Morgan, NJ Czaplewski & NB Simmons, 2017, unpublished data; GS Morgan & NJ Czaplewski, 2017, unpublished data) . The opening of the vestibular aqueduct is relatively large and slitlike, with slight breakage around the bony rim. The vestibular aqueduct abruptly tapers funnel-like to a thin duct within the dorsal bone of the common crus.

There is an obvious external swelling over the ampulla of the anterior semicircular canal, partly breached by breakage along the prefacial commissure. On the crista parotica the anterior petrosal process is broken off at its base, but the base indicates that the process was thin and flattened. The anterior petrosal process is separated anteriorly from the prefacial commissure by a narrow U-shaped notch. Immediately posterior to the anterior petrosal process, the medial wall of the fossa incudis occurs as a small, smooth-walled indentation, tapering posteriorly and broken along its ventral edge exposing more of the facial sulcus (semicanal for the facial nerve). An overhanging sharp-edged small ridge constricts the posterior end of the fossa incudis, and a small oval depression occurs in its anterodorsal portion that accommodates the crus breve of the incus. The fossa for the stapedius muscle is large and broad. The fenestra cochleae is larger than the fenestra vestibuli. The fenestra cochleae is much wider than high. The cochlear canaliculus is smaller than the fenestra vestibuli. There is a fairly prominent groove for the stapedial artery on the posteroventral lip of the fenestra vestibuli. There is a small tubercle (Fig. 1I, J, K, labeled "tu") ventral to the fenestra cochleae as seen in several vespertilionids, in *Miniopterus*, and also resembling that present in a fossil emballonurid from the late Oligocene-early Miocene of Florida (GS Morgan & NJ Czaplewski, 2017, unpublished data)).

The distal humerus ETMNH 14022 represents a moderately large vespertilionid (Figs. 2A–2E). The preserved portion of the shaft is nearly straight in anterior view, while the distal end curves gently forward in lateral view. Posteriorly the bone has a relatively deep olecranon fossa (for a bat, in which the olecranon fossa is often absent), while anteriorly it bears a deep, broad coronoid fossa and smaller confluent radial fossa. The ridges of the capitulum are aligned with the long axis of the shaft. The distal articular surface is only slightly offset laterally relative to the shaft, as in most vespertilionids, in that a line drawn to extend distally along the lateral edge of the humeral shaft follows the capitular groove and the entire lateral ridge of the capitulum with capitular tail occurs laterally beyond this line, while a similar line along the medial edge of the shaft follows approximately the medial edge of the trochlea. The trochlea and capitulum are separated by a shallow groove. There is a more pronounced groove between medial and lateral ridges of the capitulum. The spinous process of the medial epicondyle does not extend distally as far as the distal ridge of the trochlea; a small flat to slightly concave distal tip blunts the end of the spinous process. On the medial epicondyle, in medial view, a tiny depression separates the spinous process from a small, more proximally situated, medial process. Laterally there is a deep supra-epicondylar groove between the lateral epicondyle and the capitular tail. The fossae at the medial and lateral ends of the epicondyles are deep.

The proximal radius ETMNH 9755 is typical of that of vespertilionids (Figs. 2F–2J). On the posterior surface it has two small facets for articulation with the proximal ulna, an oval central one and a separate smaller, curved, medial one. The proximal articular surface shared with the humerus is dominated by the large and moderately deep central groove that accommodates the medial ridge of the humeral capitulum. Lateral to this groove is a small shallowly dished area that accommodates the lateral ridge of the humeral capitulum. Medial to the central groove is a facet for accommodation of the humeral trochlea. The flexor fossa for insertion of the biceps muscles is deep and narrow; although situated on

the medial side of the radius just distal to the head, it is still visible in anterior view. In medial (ventral) view the flexor fossa is closed off proximally by a thin flange of bone but is open distally. The proximal radius is very similar in morphology and size to that of extant *E. fuscus*.

## Discussion and comparisons

In morphological features of the M1, the GFS specimen differs from *Corynorhinus, Myotis, Nycticeius, Parastrellus, Perimyotis,* and *Plionycteris* in being larger and more robust and in details of cusp and crest shapes. It differs from the M1s of *Antrozous* and *Lasiurus* mainly in that the postprotocrista does not connect to the base of the metacone, and from *Simonycteris* in having the lingual cingulum absent at the base of the protocone. It is essentially identical to the M1 in modern specimens of *Eptesicus fuscus*.

The crown morphology and size of the m2s, one of which is still retained within a partial dentary with the ascending ramus, closely match the same elements in *E. fuscus*. In the edentulous partial dentary ETMNH 19286, no portion of a large canine alveolus is apparent mesially due to breakage, so it is not possible to determine whether more than one small lower premolar was present between the lower canine and the large p4 in this specimen. However, because the preserved alveolar configuration of the ramus fragment and its size and are consistent with those in dentaries of *E. fuscus*, the specimen is referred to *E.* cf. *E. fuscus* along with the teeth and tooth-bearing dentary in the GFS sample.

Petrosals of very few extinct late Paleogene and Neogene North American bats have been described in detail; they belong to families other than Vespertilionidae (e.g., GS Morgan, NJ Czaplewski & NB Simmons, 2017, unpublished data; GS Morgan & NJ Czaplewski, 2017, unpublished data). By comparison with petrosals of several families of recent bats available for comparison, the GFS petrosal is representative of the structure in certain vespertilionids. Although exhaustive comparisons were not made throughout the diverse and globally-distributed family, the GFS petrosal resembles that of many vespertilionids in having a flattened anterior petrosal process separated anteriorly from the prefacial commissure by a narrow U-shaped notch, in having the anterior ampulla visible externally as a prominent swelling, and in having the fenestra cochleae much wider than high. Some obvious differences serve to differentiate the Gray Fossil Site petrosal from petrosals of several widespread genera of vespertilionids. The fossil differs from the petrosal of *Plecotus* in lacking a completely laminated posterior semicircular canal, in having a much larger stapedial fossa, and in possessing an anterior petrosal process. It differs from *Corynorhinus* in having a less completely laminated posterior semicircular canal, in possessing an anterior petrosal process, and in having a lower fenestra cochleae relative to its width. Compared to *Barbastella* the fossil has a broader stapedial fossa, a narrow U-shaped notch immediately anterior to the anterior petrosal process instead of a broad connection of the process to the bone near the anterior ampulla, and thin rather than thick bone making up the prefacial commissure bridging the facial canal. It differs from *Antrozous* in lacking a sharp flange along the length of the anterior semicircular canal (variably developed among individuals of *Antrozous pallidus* examined) and a curved rather than nearly straight opening of the fenestra cochleae. It differs from *Lasiurus* in having a narrow U-shaped notch anterior to the
anterior petrosal process, a broader stapedial fossa, and a larger opening of the vestibular aqueduct that extends higher along the common crus. The unbroken portion of the pars cochlearis lacks evidence of the broad flange extending medially toward the basioccipital in *Nyctalus, Vespertilio*, and *Lasiurus* (some genera and species have the flange limited in extent to the medialmost portion not preserved in the fossil), and bears a rather stout tubercle ventral to the fenestra cochleae that is absent in *Nyctalus, Vespertilio*, and *Myotis*. The GFS petrosal further differs from that of *Myotis* in having a less completely laminated posterior semicircular canal, a small tubercle ventral to the fenestra cochleae, and a larger, flared opening of the vestibular aqueduct extending higher along the common crus. It differs from the petrosal of *Lasionycteris* in having an open (unlaminated) posterior semicircular canal. It differs from that of *Perimyotis* in having a less completely laminated posterior semicircular canal and a small tubercle ventral to the fenestra cochleae. It differs from the petrosal of *Nycticeius* in having a less completely laminated posterior semicircular canal and a long anterior petrosal process. It differs from that of *Pipistrellus* in having a tubercle ventral to the fenestra cochleae and a less completely laminated posterior semicircular canal. In morphology the petrosal is a close match with petrosals of *E. fuscus*, and the size also matches that of *E. fuscus*. Without comprehensive samples of modern bat petrosals for comparisons, and without assessments of petrosal variation within and among genera and species, it is difficult to judge individual variability in this element and provide a precise identification of ETMNH 9654. Because of its close similarity to the petrosal of modern *E. fuscus,* and in light of the relative abundance of other craniodental fossils referred to *Eptesicus* herein, this petrosal is tentatively referred to the same taxon.

Few Neogene bats are known in North America by fossil of their humeri. *Lawrence (1943)* described two genera, *Miomyotis* and *Suaptenos* based solely on nearly complete humeri from the early Miocene Thomas Farm locality in Florida. Another Thomas Farm bat, *Karstala*, is also represented by fossils of its humerus (*Czaplewski & Morgan, 2000*). *Lawrence (1943)* considered both *Miomyotis* and *Suaptenos* to be most closely related to *Myotis*. The late Oligocene genus *Oligomyotis* also was established solely upon a distal portion of humerus from Colorado (*Galbreath, 1962*), although the holotype and only known specimen is lost (*Czaplewski, Bailey & Corner, 1999*; *Czaplewski, Morgan & McLeod, 2008*). The olecranon fossa is absent or weakly developed in most western hemisphere genera of Vespertilionidae; it is best developed in *Lasiurus* and *Eptesicus*. In available characters, the distal humerus from GFS differs most notably from that of most North American bats, including *Miomyotis, Suaptenos*, *Karstala,* and *Oligomyotis*, in having a relatively well developed olecranon fossa. Lasiurine bats, known in the fossil state in North America back to the late Miocene (Clarendonian land mammal age; (*Czaplewski, Bailey & Corner, 1999*; *Czaplewski, Morgan & McLeod, 2008*)), have an even better developed olecranon fossa than *Eptesicus* and the GFS fossil. As in the available jaws and teeth, the GFS distal humerus most closely resembles that of *Eptesicus fuscus*.

*Felten, Helfricht & Storch (1973)* distinguished European species of *Eptesicus* from other European vespertilionids in having the distal humerus with a spinous process that does not extend beyond the trochlea, a transition between the trochlea and medial ridge of the capitulum on the joint surface that is concave proximally, and the proximal tip of the

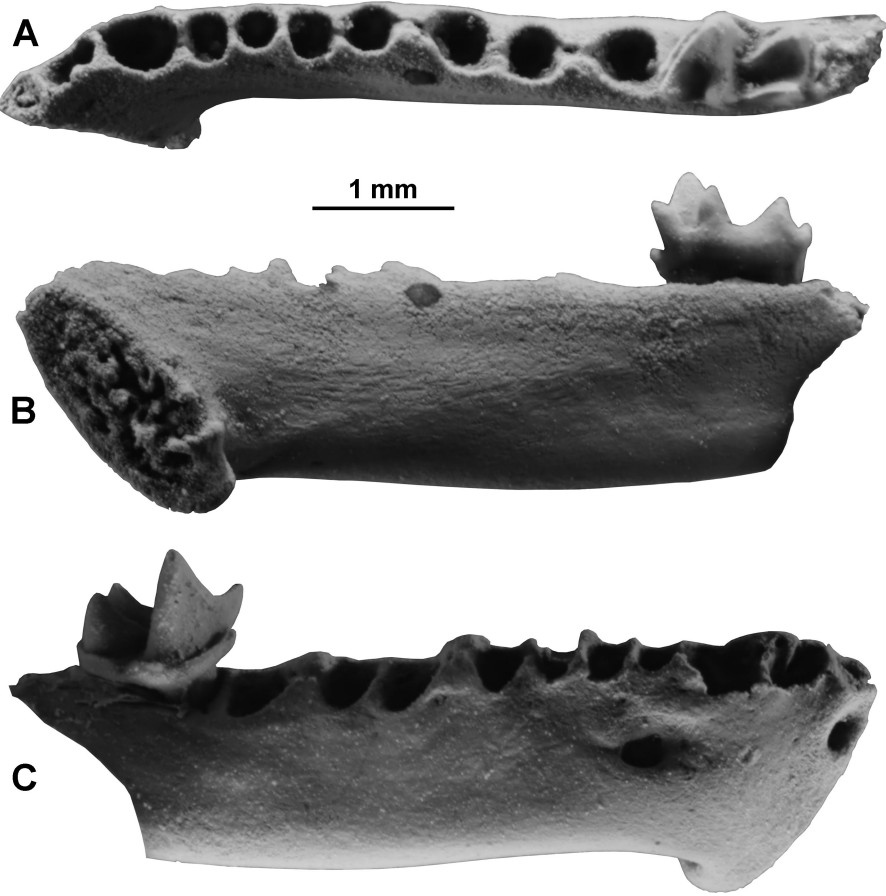

**Figure 3** **Vespertilionidae, genus indeterminate, right horizontal ramus with m3 (ETMNH 19285) from the Gray Fossil Site,** Tennessee, in occlusal (A), lingual (B), and labial (C) views. Specimen is coated with ammonium chloride for photography.

trochlea reaches the contour of the epiphysis in anterior view. In the GFS humerus the first two of these characters are met, but the proximal tip of the trochlea does not reach the outline of the epiphysis. The bone is virtually identical to the humerus of modern *Eptesicus fuscus* in details of shape, except that the notch between the spinous process and distal edge of the trochlea is deeper and more distinct. However, this feature can be individually variable: one of four specimens of modern *E. fuscus* examined had a shallower notch whereas the other three had no notch in this area. Alternatively, if additional fossils are eventually found that consistently bear the distal notch, they might help determine whether the Hemphillian GFS bat is a distinct species of *Eptesicus*.

Genus and species indeterminate
(Fig. 3)

Material: ETMNH 19285, right dentary horizontal ramus with m3 and alveoli for all of the other lower teeth.

Measurements (in mm): The m3 measures 1.2 mm in anteroposterior length, 0.7 mm in trigonid width, and 0.6 mm in talonid width. Dentary depth below the anterior alveolus of m1 is 1.25 mm. Alveolar length from c1-m3 is 5.0 mm.

Description: The ramus is from a small vesper bat, much smaller than *E. fuscus*. It indicates an animal near the size of the extant eastern North American species *Lasiurus borealis*. The alveolar formula indicates 3i, 1c, 2 or 3 p, 3m. The canine alveolus is relatively small. The three lower premolar alveoli are subequal in size, with the first and third equal and the middle one slightly smaller. For the lower premolars these three alveoli ostensibly do not allow a determination of the presence of a single-rooted p3 and double-rooted p4 or of three single-rooted lower premolars p2, p3, p4. However, in bats with three single-rooted premolars, the p3 root is typically smallest, that of p2 larger, and that of p4 largest, usually much larger than that of the adjacent p3, whereas in bats with a two-rooted p4 the roots are nearly equal in size. The m3 is relatively unreduced with low trigonid cusps and a narrower but relatively wide, basined talonid that retains three distinct and well-developed cusps, hypoconid, entoconid, and hypoconulid. The m3 postcristid exhibits clear myotodonty.

Discussion and Comparisons: Among recent vespertilionid genera with three root sockets between the lower canine and the first lower molar, there are two possible premolar configurations, a single-rooted p3 and double-rooted p4 (1 + 2), or single-rooted p2, p3, and p4 (1 + 1 + 1). The premolar alveolar count of 1 + 2 is known in *Antrozous, Barbastella, Bauerus, Eptesicus, Histiotus, Ia,* some *Lasiurus, Nycticeius, Otonycteris, Perimyotis, Parastrellus, Rhogeessa, Scotomanes*, and *Vespertilio*, whereas the 1 + 1 + 1 count occurs in *Corynorhinus* and *Idionycteris*. Among northern hemisphere Neogene bats of North America and Eurasia, ETMNH 19285 differs from *Ancenycteris, Hanakia, Eptenonnus, Quinetia*, and *Submyotodon*, and from the extant species (some of which are also known as Quaternary fossils) of *Lasionycteris, Myotis,* some *Lasiurus, Plecotus, Euderma, Scotoecus,* and *Scotozous* in having three lower premolar alveoli. ETMNH 19285 differs from *Barbastella, Nyctalus, Scotoecus*, and *Scotozous* in showing myotodonty rather than nyctalodonty of the molar postcristid. It differs from *Otonycteris* and *Scotomanes* in having m3 with a well-developed rather than greatly reduced m3 talonid. ETMNH 19285 differs from *Karstala, Samonycteris, Idionycteris, Histiotus, Ia, Antrozous, Bauerus, Otonycteris, Nyctalus, Scotoecus, Scotomanes,* and *Scotozous* in its much smaller size, and from *Parastrellus* in its much larger size (although the m3 has about the same dimensions in both the fossil and *Parastrellus hesperus, Parastrellus* has the alveolar tooth row length about 5.8 instead of 5.0 mm). Compared to *Corynorhinus* and *Idionycteris,* premolar alveoli in the GFS fossil are more nearly equal to one another in size rather than having the p4 alveolus larger than those of p2 and p3. Moreover, the m3 trigonid cusps–especially the protoconid–are distinctly shorter than in *Corynorhinus*, and the m3 hypoconulid is better developed. Compared to *Lasiurus ( subgenus Lasiurus), Lasiurus (Dasypterus), Rhogeessa*, and *Scotomanes,* the GFS fossil differs in having a stronger talonid on m3 with better developed hypoconulid and hypoconid. It has a deeper, more robust horizontal ramus and more robust teeth than *Perimyotis subflavus*. Comparisons cannot be made with the Neogene vespertilionids *Paleptesicus* (includes *P. priscus* only; see (*Horácek, 2001*)), *Plionycteris, Potamonycteris, Simonycteris, Suaptenos*, and *Miomyotis* because of

non-comparability of available skeletal elements. No distinguishing features of the GFS fossil could be found in comparisons with *Nycticeius*, the smaller species of *Eptesicus* (i.e., numerous small Palearctic, Neotropical, Afrotropical, and Indomalayan species), *Nycticeinops*, *Vespertilio*, *Miostrellus*, and *Pipistrellus*. Because these genera cannot be distinguished from one another based on the morphological features preserved in ETMNH 19285, it is not possible to assign the GFS specimen to a genus.

## DISCUSSION

Fossils of *Eptesicus* of several species have been described from the Miocene of Europe (*Storch, 1999*; *Rosina & Sinitsa, 2014*) and from the early Pliocene to Pleistocene in Africa (*Gunnell, 2010*). In North America, Cf. *Eptesicus* sp. (a quite small vespertilionid) occurred in the late Miocene (late Hemphillian land mammal age) of the Redington fauna, Arizona (*Czaplewski, 1993*), while *E. ?fuscus* was listed in the early Pliocene (early Blancan land mammal age) at Beck Ranch, Texas (*Dalquest, 1978*), *Eptesicus* sp. in the late Pliocene (late Blancan) at Inglis 1A, Florida (*Morgan, 1991*), and *Eptesicus fuscus* in the Pliocene (Blancan) in San Bernardino County, California (*Czaplewski, 1993*). The late Miocene species *Eptesicus* "*hemphillensis*" from Coffee Ranch, Texas (middle Hemphillian; *Dalquest, 1983*) is taxonomically invalid, although another specimen from Coffee Ranch is inseparable from *E. fuscus* (see *Czaplewski, Morgan & McLeod, 2008*). *Eptesicus* species are also known from the Pleistocene of South America.

The species *Eptesicus fuscus* has a relatively extensive Pleistocene record and distribution covering much of its modern North American range east of the Rocky Mountains and in Mexico (*Kurta & Baker, 1990*; *Faunmap Working Group, 1994*; *Arroyo-Cabrales, 2005*), in the Quaternary in parts of northern South America within and beyond its present-day range there (*Linares, 1968*; *Czaplewski & Cartelle, 1998*; *Lessa, Cartelle & de Aguiar, 2005*; *Rodrigues & Ferigolo, 2005*), and in the Caribbean (*Morgan, 2001*). The genus *Eptesicus* in the present day is widely distributed in the northern hemisphere (Holarctic biogeographic regions) as far north as the Arctic Circle, and southward through many parts of Eurasia and Africa. In the Neotropical region it extends southward through Central America and South America, south to central Argentina, as well as in northern and southern Africa, and southeastern Asia (northern Indomalayan region). The genus has considerable recent diversity and contains about 25 extant species of small to large body size for the family (*Corbet, 1978*; *Corbet & Hill, 1992*; *Reid, 1997*; *Simmons, 2005*; *Gardner, 2007*; *Happold & Happold, 2013*; *Juste, Benda & Ibáñez, 2013*).

The living, widespread Eurasian species *E. serotinus* is very similar in morphology to the North American *E. fuscus*, and the two species (and possibly other species of the genus) cannot be distinguished by morphological characteristics of skeletal elements. At one time *Koopman (1994)* synonymized the two species as *E. serotinus*, but later authors reversed this designation (*Simmons, 2005*). The morphological similarity of the GFS fossils with *E. fuscus* and *E. serotinus* could point to the immigration from Europe or Asia of an ancestral *Eptesicus*, but molecular phylogenetic studies show the two species to have separate origins (*Roehrs, Lack & Van Den Bussche, 2010*; *Agnarsson et al., 2011*; *Juste, Benda*

& Ibáñez, 2013) and an autochthonous origin in North America is likely. For this reason, and because of the very close morphological correspondence to recent *E. fuscus*, I tentatively refer the GFS fossil to *E.* cf. *E. fuscus*.

In any case, the few GFS fossils cannot address these questions unless better more complete specimens are found. A Miocene biogeographic connection for tropical forest between eastern Asia and western North America via Beringia is already well established (*Wolfe, 1994a*; *Wolfe, 1994b*; *Sirkin & Owens, 1998*; *Reinink-Smith & Leopold, 2005*). The fossil occurrence at GFS of several vines including *Sinomenium* and several species of grapes (*Vitis*) having eastern Asian affinities (*Gong, Karsai & Liu, 2010*; *Liu & Jacques, 2010*), as well as the GFS fossil mammals *Pristinailurus* and *Arctomeles,* related to the red panda and Eurasian badger (*Wallace & Wang, 2004*), respectively, suggest that perhaps the ancestry of *E. fuscus* should also be looked for in eastern Asia.

This report includes the first records of bats from the Gray Fossil Site, and the second record of *Eptesicus* in the late Neogene of eastern North America, the other being the Florida Pliocene record. Recent bats of the genus *Eptesicus* occur on six continents and are tolerant of a wide range of habitats and environmental conditions from lowlands to highlands and rain forest to desert (*Emmons & Feer, 1997*; *Reid, 1997*; *Wilson & Ruff, 1999*; *Gardner, 2007*; *Happold & Happold, 2013*; *Ceballos, 2014*). For example, *Eptesicus fuscus* itself is widespread in the extant biota of northwestern South America, Central America, many Caribbean Islands, and North America throughout the United States and much of southern Canada. It is a habitat generalist and utilizes a variety of habitats across this broad range (*Agosta, 2002*); foraging habitat is in relatively open vegetation, woodlands, forest, and forest clearings (at higher elevations in tropical mountains, lower in the temperate zone) and roosting (including hibernation in the coldest season) occurs in caves, rock crevices, tree hollows, and human-built structures (*Linares, 1968*; *Harvey, Altenbach & Best, 2011*). As a result, the GFS *E.* cf. *E. fuscus* and the second unidentified vespertilionid cannot provide new information relevant to paleoenvironmental interpretations of the GFS; nevertheless, they are consistent with earlier interpretations of subtropical oak-hickory forest for the area in the late Hemphillian. Stable isotopes recovered from GFS browsing mammals suggest that in the late Miocene-early Pliocene there was minimal seasonal variation in temperature and precipitation at GFS (*DeSantis & Wallace, 2008*; *DeSantis & Wallace, 2011*). Together with annual low temperatures probably above 5.5 °C (*Shunk, 2011*) and a predicted minimum average annual temperature of at least 22 °C (*Mead et al., 2012*), these paleoenvironmental conditions suggest that hibernation in the area surrounding GFS and southern Appalachian Mountains might not have been necessary or possible for the bats.

## ACKNOWLEDGEMENTS

For the invitation to study bat specimens from the Gray Fossil Site and for logistical support and encouragement I thank Jim I. Mead, Blaine W. Schubert, Sandy Swift, and Steven C. Wallace. For the loan of specimens and aid at the GFS and in the ETMNH collection I am grateful to SC Wallace and April S. Nye. I thank Brandi Coyner and Janet Braun of the

Oklahoma Museum of Natural History for the loan of recent mammal skeletal material. Steve Westrop allowed use of his bellows camera, Stackshot rig, and stacking software; Roger Burkhalter helped with photographing specimens and rendering images. I thank Ivan Horácek and Gregg Gunnell for providing helpful reviews of the manuscript.

### Funding

The specimens involved in the study came from sediments that were dug, screenwashed, and sorted by many workers and volunteers under the supervision of SC Wallace, and supported by NSF grant 0958985 to Wallace and BW Schubert. Funding for the study of the specimens by the author was provided by the Sam Noble Oklahoma Museum of Natural History. The funders had no role in study design, data collection and analysis, decision to publish, or preparation of the manuscript.

### Grant Disclosures

The following grant information was disclosed by the author:
NSF: 0958985.
Sam Noble Oklahoma Museum of Natural History.

### Competing Interests

The author declares that he has no competing interests.

### Author Contributions

- Nicholas J. Czaplewski analyzed the data, contributed reagents/materials/analysis tools, wrote the paper, prepared figures and/or tables, reviewed drafts of the paper.

### Data Availability

The raw data are the taxonomic identifications of specimens, and the measurements and morphological descriptions on which they are based are in the Systematic Paleontology section of the manuscript.

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
