# Peer review of "First report of bats (Mammalia: Chiroptera) from the Gray Fossil Site (late Miocene or early Pliocene), Tennessee, USA"

_PeerJ, doi:10.7717/peerj.3263_

## Round 0.1 · original submission · Minor Revisions

Dear Dr Czaplewski

You Ms # 15937 entitled "First report of bats (Mammalia: Chiroptera) from the Gray Fossil Site (late Miocene or early Pliocene), Tennessee, USA " which you submitted to PeerJ has been reviewed by two reviewers and the editor.

Both reviewers are coincident that your contribution is worth for publication and should be accepted after minor revision, concerning mainly misspellings, missing references, and some rewording, among others. Please pay attention particularly to suggestions made by Reviewer #2. After this, I think your Ms would be acceptable for publication.

So, I am requesting that you revise the minor suggestions mentioned above and resubmit your Ms to PeerJ.

Looking forward to receiving your revision.

Sincerely, Claudia Marsicano

·

Basic reporting

A well organized professional paper by one of leading personalities of the branch. All basic information on described specimens are presented including high quality photographs.

Experimental design

The comparisons confront the characters significat in taxonomic and phylogenetic respects. Stratigraphical and paleoecological setting of the deposits are explained with relevant details.

Validity of the findings

Presence of Eptesicus cf. fuscus in the late Miocene site in Tenneessee supplements the former fossil record of the species in a relevant way. The comparison of the cochlear characters is impressive.
In my eyes, the specimen ETMNH 19285 (dentary with m3 and alveoli for
other lower teeth) is the most exciting item of the contribution. It is undoubtedly a vespertilionid of Pipistrellus-like appearance, yet myotodont with a large p2 and unreduced m3 with particularly long talonid almost without any hypoconulid. Such a combination is relatively rare in vespertilionid bats but characterizes well the North American genus Parastrellus (comp. Hoofer et al. 2006). The differences in size not in proportion of particular teeth would support a difference between the fossil form and extant P.hesperus at, say, a species level. Of course, it is upon the author whether would find such a possibility relevant - a detailed comparison with extant hesperus would be needed.

row 305 replace 12.5 mm with 1.25 mm

Additional comments

An interesting paper worth of publication.

·

Basic reporting

This is a well written paper that is concise and without need of much change. References are complete, the figures are okay (I think it would benefit the paper to have Micro-Ct scans of the petrosal described here in order to better see the detail but this is not a requirement). Results are carefully presented and not over-stated.

Experimental design

Not really applicable other than the comparative anatomy which is thorough

Validity of the findings

The taxonomic findings are conservative as they should be based on the material at hand - habitat reconstructions based on analogy with living Eptesicus are not contradictory to those based on other lines of evidence.

Additional comments

I think this is a fine paper, more or less as is - I had only a few minor suggestions that I will detail below.

The only relatively major remark I have is on the discussion of the petrosal - as the author rightly points out, very little work has been done documenting petrosal morphology in vespertilionids or for most other bats for that matter. I found the description relatively difficult to follow based on the illustrations presented - additionally I think the detailed comparative anatomy presented in the discussion section goes far beyond what most readers would be able to discern or understand. It seems clear that this author is working on a more in-depth analysis of bat petrosal anatomy (citing two in prep/review papers) and thus clearly grasps this topic better than most would. Perhaps a scaled down version of the petrosal section would be warranted for this paper since the details of petrosal anatomy across bats has not yet been presented elsewhere.

Minor Items:

1) I found the wording of first sentence of the abstract to be very confusing - perhaps it would be best to break this down into two sentences (for example the first line mentions "vertebrate fossils" and then the second line mentions "eight fossils" - no need to repeat "fossils" twice)

2) line 43 - should this read "ponds and small streams" instead

3) line 67 - in Methods section - some short mention should be made in this section about upper teeth being designated with an upper case letter and lower teeth with a lower case letter

4) line 148 - funnel-like (add hyphen)

5) line 380 - I think adding some citations for the geographic range and taxonomic makeup of Eptesicus would be appropriate here

6) line 391 - I think it might be good to begin a new paragraph starting with "In any case,"

7) line 405 - please add in some citations to support the habitat range of Eptesicus

8) line 608 - the Wallace et al. reference is incomplete (no publication information included)

That's all - I think the illustrations are adequate although as mentioned above a Micro-Ct scan of the petrosal would be much better in order to improve the visualization the details discussed - hopefully when the papers the author cited as in prep/press wherein more broad comparisons are published, they will be documented by the use of Micro-Ct....

---

## Round 0.2 · accepted · Accept

Dear Dr Czaplewski

It is a pleasure to accept your Ms # 15937 entitled "First report of bats (Mammalia: Chiroptera) from the Gray Fossil Site (late Miocene or early Pliocene), Tennessee, USA " which you submitted to PeerJ.

Thank you for your fine contribution. We look forward to your future contributions to the Journal.

cheers, Claudia Marsicano